# Evaluation and Predictive Factors of Complete Response in Rectal Cancer after Neoadjuvant Chemoradiation Therapy

**DOI:** 10.3390/medicina57101044

**Published:** 2021-09-30

**Authors:** Linda Kokaine, Andris Gardovskis, Jānis Gardovskis

**Affiliations:** 1Department of Surgery, Riga Stradins University, Dzirciema Street 16, LV-1007 Riga, Latvia; andris.gardovskis@rsu.lv or; 2Pauls Stradins Clinical University Hospital, Pilsoņu Street 13, LV-1002 Riga, Latvia

**Keywords:** locally advanced rectal cancer, tumor response, predictive factors of complete response, neoadjuvant therapy

## Abstract

The response to neoadjuvant chemoradiation therapy is an important prognostic factor for locally advanced rectal cancer. Although the majority of the patients after neoadjuvant therapy are referred to following surgery, the clinical data show that complete clinical or pathological response is found in a significant proportion of the patients. Diagnostic accuracy of confirming the complete response has a crucial role in further management of a rectal cancer patient. As the rate of clinical complete response, unfortunately, is not always consistent with pathological complete response, accurate diagnostic parameters and predictive markers of tumor response may help to guide more personalized treatment strategies and identify potential candidates for nonoperative management more safely. The management of complete response demands interdisciplinary collaboration including oncologists, radiotherapists, radiologists, pathologists, endoscopists and surgeons, because the absence of a multidisciplinary approach may compromise the oncological outcome. Prediction and improvement of rectal cancer response to neoadjuvant therapy is still an active and challenging field of further research. This literature review is summarizing the main, currently known clinical information about the complete response that could be useful in case if encountering such condition in rectal cancer patients after neoadjuvant chemoradiation therapy, using as a source PubMed publications from 2010–2021 matching the search terms “rectal cancer”, “neoadjuvant therapy” and “response”.

## 1. Introduction

Rectal cancer represents approximately 35% of the total colorectal cancer incidence [1]. This is a heterogeneous type of cancer that attracts clinical attention due to the variety of treatment options. Most of the patients with early rectal cancer can be managed by surgery alone, however a significant proportion of patients present with a locally advanced disease that demands neoadjuvant therapy (NAT) with a purpose to reduce the local tumor burden and increase the safety and efficacy of further surgical treatment [2]. NAT involves a variety of treatment options including radiotherapy and chemotherapy, used alone or in combination [1].

Tumor response to NAT is an important prognostic factor for locally advanced rectal cancer (LARC) [3]. Following NAT, a clinical complete response (cCR) can be obtained in 10–40% of rectal cancer patients, however, it should be noted that the numbers of real pathological complete responses (pCR) are on average two times less [1,2]. Patients with a pCR to NAT have lower rates of local recurrence, improved survival as compared to patients who don’t achieve pCR. The 5-year recurrence-free survival rates are 90.5%, 78.7% and 58.5% for patients with complete, intermediate and poor response [4]. Moreover, patients with pCR after neoadjuvant chemoradiotherapy (nCRT) have improved distant metastatic rates 7–10.5%, when compared to poor responders of NAT—26–31%, respectively [5,6].

The possibility to predict patient’s response to nCRT would help to guide more personalized treatment strategies [1]. This would allow a more precise selection of patients who would benefit the most from nCRT, protect patients from potentially unnecessary treatment, and allow identification of the best candidates for nonoperative management. Unfortunately, at present, there are no certain preoperative predictive factors that could determine the response to the NAT and could be implemented into clinical practice so far [7].

## 2. Tumor Reaction to the NAT—Defining the Terms

Tumor reaction to the NAT is mainly described with the following terms–“downstaging”, “downsizing”, “downshifting”, “regression” and “response” [8,9].

Tumor downstaging–characterizing the changes in tumor stage classification (0, I, II, III and IV), describing the change from a higher stage group to a lower one (e.g., from stage III to stage II).

“Tumor downshifting”–characterizing the changes in tumor extent (T1, T2, T3, T4), lymph node involvement (N0, N1, N2) or presence of distant metastases (M0, M1), describing the change from a higher extent to a lower one (e.g., from T3 to T2).

“Tumor downsizing”–characterizing the decrease of the tumor size, but does not always mean a simultaneous “downshifting” or “downstaging” (e.g., T3 tumor may reduce in its size without reduction of the tumor extent).

“Tumor regression”–refers to the pathological ratio of residual viable tumor to scar after NAT (chemotherapy, radiotherapy or combined) which reveals nothing about the change in tumor size, nor about downstaging/shifting. Unfortunately, “tumor regression” is often used to indicate all forms of tumor response to treatment [9].

“Tumor response”–may be classified as complete, incomplete or non-response. The Response Evaluation Criteria in Solid Tumors (RECIST) guidelines were created in 2000 and later updated in 2009 (Table 1) [10].

A target lesion is defined as ≥10 mm measurable lesion present at baseline, up to a maximum of five lesions total, representative of all involved organs, based on size. Non-target lesions represent all other lesions (or sites of disease) including pathological lymph nodes [10].

The revised RECIST guidelines are currently the closest standardized and accepted response criteria available [3].

## 3. Complete Tumor Response after NAT

### 3.1. Clinical Complete Response (cCR)

In certain cases, tumors may completely respond to NAT. In general, cCR is defined as the absence of clinically detectable residual primary tumor on clinical examination and endoscopy at least 4 weeks after completion of NAT [1].

According to a recent systematic review by Dattani et al. (17 studies comprising 692 patients), the reported cCR rate was 22.4% [11]. Up to this, there have been several definitions of cCR described in studies focusing on the nonoperative management of rectal cancer [12].

Definition of cCR by Brazilian group led by Habr-Gama who has the longest experience with this population and has attempted to standardize the classification of cCR includes several clinical parameters: (1) absence of residual ulceration, (2) no mass or mucosal irregularity, (3) absence of extra-rectal disease on imaging, (4) whitening of mucosa and telangiectasia acceptable. Since a transient response is common, the authors defined initial cCR occurring on first assessment >8 weeks after nCRT, and sustained cCR for patients who maintain the response from 10 weeks until at least 12 months after nCRT [13].

Definition of cCR in the 2017 European Society for Medical Oncology (ESMO) guidelines is including several criteria:

Minimal criteria

the absence of any irregularities or a palpable tumor on digital rectal examination,no visible lesion on endoscopy with the exception of a flat scar, telangiectasia, or whitening of the mucosa;

Additional criteria

3.absence of any residual tumor in the primary site and draining lymph nodes on imaging with magnetic resonance imaging (MRI) or endorectal ultrasound (ERUS),4.negative biopsies from the scar,5.an initially raised CEA level that returns to normal (<5 ng/mL) [2].

### 3.2. Pathological Complete Response (pCR)

Pathological complete response (pCR) is defined as an absence of viable tumor cells after full pathologic examination of the resected specimen (pT0N0M0) [12]. Tumor regression grade (TRG) is a method to stratify primary tumor response to NAT based on a histopathological assessment of residual tumor cells and degree of tumor regression and replacement. Various TRG classification systems are used—Mandard (1994), Dworak (1997; modified Dworak—2003), Memorial Sloan-Kettering Cancer Center (MSKCC) classification (2008) and Ryan/American Joint Committee on Cancer (AJCC) 7th Edition (2010) (Table 2) [1,13,14,15,16]. Mandard et al. introduced a TRG system for oesophagal carcinoma, which has been applied in other digestive tract malignancies as well. In the Mandart system TRG is classified into five grades, and CR is defined as TRG1—complete regression, fibrosis extending through the different layers of the wall, no viable cancer cells. According to Dworak et al. TRG is classified into four grades, and CR is defined as TRG4—no tumor cells, only fibrotic mass or acellular mucin pools. Another classification offered by Ryan et al. and is currently advised by the American joint committee on cancer (AJCC); in this classification, TRG is classified into four grades and CR is defined as TRG0—no viable cancer cells. MSKCC is offering classification of three groups, depending on the tumor response rate; CR is defined as TRG1—100% tumor response.

Another 14 classification systems are described in the literature with different modifications of TRG—Modified Mandard (Ryan) (2005), Werner and Hoffler (2000), Cologne (2005), Bujko/Glynne Jones (2010), College of American Pathologists (2008), RCPath system (1997), RCRG system (2002), Modified RCRG system (2009), Japanese classification (2012), Ruo (2002), Junker and Muller (2003), Rodel (2005), Swellengrebel et al. (2014), Modified Mandard TRGN by Dhadda et al. (2014) [17].

Irrespective of the grading system, assessment of pathological response is based on: (1) tissues replacing tumor cells in areas where the tumor has regressed, (2) residual tumor cells. These “replacement” tissues may be inflammatory or fibrotic; they may consist of acellular mucin pools or calcifications and necrosis [14].

Analysis of the National Cancer Database in 2017 reported a pCR rate of 13% in overall patient cohort 27,532 [16]. Unfortunately, cCR and pCR are not always consistent with each other. In some studies, cCR was reported in up to 40% of the cases with NAT. After surgery, it was observed that half of these resected specimens contained tumor cells at histopathology examination [18,19]. Chari et al. found that even 22 of 43 (51%) patients who underwent NAT for rectal cancer had a cCR, however, only 11 of those 22 patients had a pCR [20]. Seong et al. reported a cCR rate of 23.8% in patients with primary unresectable rectal cancer receiving NAT, however after resection—only 9.5% had a pCR [21]. Other studies of complete clinical responders and pathologic complete responders (e.g., Issa et al., 2012, Nyasavajjala et al., 2009, Nair et al., 2008, Hiotis et al., 2002) showed that in 33–81% of cases there is a disconcordance between cCR and pCR [3]. Such a remarkable difference is possibly related with a selected group of the patients (T2–T4 rectal cancer, initial node positivity), applied therapy (type of radiation and/or chemotherapy), the used criteria for cCR detection, the sensitivity of radiological investigations, the timing of the tumor response evaluation, individual patient factors and interpretation of the specialist.

To distinguish these two groups of assessment in case of reached CR, it is offered to apply an appropriate TNM staging, respectively–ycT0N0M0 in case of cCR and ypT0N0M0 in case of pCR [8].

## 4. Diagnostic Accuracy of CR

### 4.1. Reassessment/Response Assessment after NAT

The standard methods of clinical reassessment of patients after NAT are based on clinical examinations that include digital rectal examination (DRE), proctoscopy or endoscopy and re-imaging with MRI. These findings determine appropriate surgical strategy, the type of planned operation and the possibility of applying a “watch and wait” (“W&W”) strategy. It is recommended to re-evaluate the primary tumor and potential circumferential resection margin (CRM) with MRI in LARC patients after NAT prior to resection to provide clear surgical margins. Although, it must be considered that re-imaging after NAT may both overestimate and underestimate (due to the poor discrimination between residual tumor and radiation-induced fibrosis) pathological response and tumor downstaging [2]. Part of the concern in accurate determination of CR is that DRE, biopsy and post-NAT MRI are unreliable for distinguishing fibrosis from small, residual islands of tumor tissue, indicating a low positive predictive value [3]. The assessment methods of tumor response after NAT and their predictive value are described further.

### 4.2. Digital Rectal Examination (DRE)

The objective examination is one of the most important procedures for the evaluation of tumor response and DRE is irreplaceable in this. Quite often patients with tumor regression have a reduction of symptoms. The basic criteria to consider a cCR during the DRE include the absence of: (1) any palpable mass or irregularity, (2) ulceration, (3) stenosis. The surface of the rectal wall has to be smooth and regular [22]. Although, a flat, small ulcer with smooth edges and without signs of residual exulcerated or polypoid tissue may be considered as a potential CR as well [23].

### 4.3. Proctoscopy/Endoscopy

Endoscopic evaluation of the originating area of the initial tumor is a persistent key component of clinical assessment. Most of the applied “W&W” strategies have included DRE and proctoscopy/endoscopy as part of the reassessment protocol. During the procedure, it is important to seek for any superficial ulcer or irregularity missed during DRE. Telangiectasia or a flat, white scar and loss of rectal wall elasticity are frequent endoscopic findings among patients with cCR. Possible signs of incomplete response include the presence of: (1) any residual ulcer (deep or superficial, with or without necrotic center), (2) significant stenosis of the lumen. Although flexible endoscopes may provide visual documentation of clinical response, rigid proctoscopy may be an adequate extent for the majority of patients. Overall, endoscopy is a helpful tool, but it is not sufficient on its own. A weak point of endoscopy is that it provides information only about the luminal side of rectum and nothing about the deeper layers and the mesorectum. As more remarkable fibrotic changes during examination could be easily interpreted as a residual disease at the site of the tumor, MRI can provide this additional information, that can be crucial for further decision making [22,23,24].

### 4.4. Biopsy

In case of detected cCR in DRE and proctoscopy/endoscopy, additional biopsies are not recommended. Even in the situation of incomplete clinical response, the biopsy results should be interpreted with caution. The reported negative predictive values of these biopsies among patients with significant response are low. Therefore, a negative biopsy in the condition of incomplete clinical response does not exclude the presence of microscopic residual cancer. The Sao Paulo group in 2012 did a retrospective comparative study to find out the value of post-NAT biopsies and it was concluded that the sensitivity was 50% and the negative predictive value—11%, which might be explained by the geographical miss of the biopsy [25]. Several studies have tried to detect the presence of residual tumor with post-NAT biopsies and the sensitivity of the examination was rated from 19.4–69.4% (Perez et al., 2014–69.4%; Xiao et al., 2016–19.4%; Kuo et al., 2012–34.7%; Lopez-Lopez et al., 2016–65.3%) [26]. Hayden et al. in 2012 introduced the concept of “tumor scatter” defined as the (1) microscopic tumor cells present in the absence of a visible ulcer or (2) presence of cancer cells outside of the visible ulcer. In this study, the residual tumor was identified outside the visible ulcer or in the absence of an ulcer in 49% of cases and some of the samples demonstrated the presence of the residual tumor cells up to 4 cm away from the primary tumor bed [27].

One of the problems of downstaging/downshifting is the lack of homogeneity in response options–there is significant heterogeneity in reaction to treatment even within one tumor. This causes considerable difficulties in staging tumors with and without NAT: after NAT, tumor can remain present in the outer layers of the rectal wall, while disappearing in the inner layers, as it has been concluded in two independent studies [28,29].

Gosens et al. in 2007 raised an idea that there are at least two ways how rectal cancer can respond to NAT: shrinkage or fragmentation. Shrinkage (reduction in the direction of the mucosa) is the preferred way of response, often leading to downshifting of T stage, allowing adequate monitoring with endoscopy or imaging and considering organ preserving surgery or “W&W” strategy. Fragmentation (the destruction of the main tumor mass and formation of small groups of tumor cells) is a condition more difficult to deal with—the limited size of the tumor fragments is below the ability of resolution on imaging and might lead to unradical surgical treatment or unfounded “W&W” strategy, resulting in local recurrence [26].

Three recent studies (Perez et al., 2014; Hav et al., 2015; Fernandez-Acenero et al., 2017–comprising 289 patients in total with LARC) showed that fragmentation was present in 40% of cases, Smith et al. in 2014 in a study of 45 cT3 patients showed a fragmented response in 80% of cases [26,30,31,32,33]. Fragmentation or scattering of tumor cells causes also a clear inconsistency between macroscopy and microscopy. Tumor fragmentation is associated with less downstaging, more frequent residual lymph node metastases, positive resection margins in case of surgery and an overall poor outcome [26,27]. If fragmentation is considered to be one of the response mechanisms, then this explains not only the limited value of post-NAT biopsies but as well–higher recurrence rates in case of advanced tumors during the “W&W” strategy [26,34].

Another important fact to be noted–not all patients with clinically confirmed CR after NAT have concurrent complete regression of lymph node metastases. The presence of remaining a tumor in lymph nodes after a CR of the primary tumor is estimated to be found in 7% of cases, demonstrating the diversity of therapy response [26,35]. This probability seriously threatens the oncological safety in case of organ sparing surgery after NAT and the application of the “W&W” strategy. The risk of lymph node positivity is dependent on several factors related to tumor biology, including differentiation grade, histological type, interaction with the microenvironment (i.e., tumor budding) and invasion of lymphatic vessels [36].

### 4.5. Magnetic Resonance Imaging (MRI)

The accuracy of rectal cancer staging at MRI in post-NAT tumors is lower if compared with a primary staging of rectal cancer at MRI [37].

To standardize the post-NAT imaging approach, MR-modified Mandard grading system (mrTRG) has been used for the last decade to identify the post-therapy changes (fibrosis/residual tumor) and define the responder groups depending on the qualitative changes within the treated tumor [38]. Application of mrTRG can determine good and poor responders. It does not correlate directly with histopathological TRG, and there is a discrepancy with RECIST tumor measurements [2].

According to the mrTRG grading system patients are categorized in 5 groups: TRG1—complete radiologic response, TRG2—good response, TRG3—moderate response, TRG4—slight response, TRG5—no response [38,39].

Scoring systems established by the European Society of Gastrointestinal and Abdominal Radiology (ESGAR) and the Magnetic Resonance Imaging and Rectal Cancer European Equivalence Study (MERCURY) study group are mostly similar but there are some differences in their definitions of good responders (Table 3) [40].

A recent analysis by Seo et al. in 2019 claimed that mrTRG is ten times more likely to identify patients with pCR if compared with isolated clinical assessment, especially in case of residual mucosal abnormality [41].

An analysis of 33 studies reported that on post-NAT restaging with MRI using standard T2 weighted sequences the overall sensitivity and specificity was 50% and 91%. The relatively low sensitivity is related to the fact that conventional MRI cannot differentiate between fibrosis and a tumor.

Some studies show that sensitivity of the examination can be notably improved after adding diffusion-weighted imaging (DWI) but without any particular effect on specificity [42]. The apparent diffusion coefficient (ADC) measured via DWI-MRI is a typical radiological marker of the nCRT response of rectal cancer [43]. As diffusion properties of water molecules may differ in regions of high cellularity (often observed within tumor tissues), tissue necrosis or fibrosis, it may be used as a helpful tool to evaluate tumor response to nCRT [44]. ADC quantifies such restriction to diffusion of water molecules (observed as high signal intensity in the region of the previous tumor) and it has been associated with CR [45].

Lambregts et al. in 2011 studied total 120 patients with LARC who underwent CRT and post-NAT evaluation by standard T2 weighted MRI and DWI for identification of complete responders. The sensitivity ranged from 0–40% on standard MRI vs. 52–64% after the addition of DWI. It was concluded that DWI-MRI could significantly improve not only the sensitivity for identification of complete responders but also the specificity which was found to be greater than 90%, subsequently, the risk for underestimation of the residual tumor could be reduced to <10% [43]. Although, Maastrich group work in 2015 showed that radiological assessment with additional DWI missed 15% of patients with pCR [23].

Another commonly investigated functional MRI technique is dynamic contrast-enhanced (DCE) or “perfusion” MRI. DCE-MRI can obtain quantitative parameters related to microvascularity and tissue perfusion by measuring the inflow of intravenously injected contrast agents and the leakage of contrast into the extracellular space, which have shown significant correlations with tumor response. Although DCE-MRI is already routinely applied in breast and prostate cancer imaging, in case of rectal cancer it has been applied only under research conditions [45].

Less frequently studied techniques include MR spectroscopy, blood oxygenation level-dependent (BOLD) MR and magnetization transfer (MT) imaging. Improved methods of DWI acquisition such as diffusion kurtosis imaging (DKI) and intravoxel incoherent motion (IVIM) imaging have also been described [45].

### 4.6. Computed Tomography (CT)

CT is not applied for evaluation of the local response of rectal cancer after nCRT due to its relatively lower sensitivity and specificity if compared with MRI. Due to the poor soft-tissue contrast, the accuracy of standard CT scan in evaluation and prediction of pCR, has been estimated at less than 50% [46].

CT is used only for pre-treatment radiological TNM staging and routine restaging of the chest and abdomen is not recommended even after nCRT. Only patients with initially more advanced cancers (cT4, cN2), the presence of extramural venous invasion (EMVI) and threatened CRM should be re-staged within 3 months from primary staging to exclude progressive metastatic disease [2].

### 4.7. Positron Emission Tomography/Computed Tomography (PET/CT)

Radiologic investigations that include additional metabolic information in comparison with standard radiologic imaging are expected to improve the overall accuracy of complete tumor response identification. PET-CT imaging has been evaluated for the prediction of response to NAT, although the role of this investigation has not been clearly defined yet [3].

The application of molecular imaging may add information to standard structural/anatomical features and provide better differentiation of fibrosis and residual tumor. The use of Fluorine-18 2-fluoro-2-deoxy-D-glucose (FDG) allows to evaluate tissue metabolism (standardized uptake values–SUV) of essential regions, and fused images of PET and CT may point out residual cancer tissue [47].

For the detection of residual cancer tissue, PET-CT findings present sensitivity of 93% and specificity of 53% and overall accuracy of 85% [3]. The accuracy of PET-CT in the prediction of cCR reaches 96% if it is combined with clinical evaluation [48].

Considering the fact that a metabolic CR is not equivalent to pCR in all cases, several aspects are being assessed to increase the value of PET-CT (e.g., the metabolic tumor volume; the total lesion glycolysis (TLG) and change of its percentage; a reduction in the maximal SUV by applying variable interval periods) [7,48].

A study by Perez et al. in 2012 has recommended the combination of volumetric reduction in tumors and SUV variation to predict CR to NAT. Application of individual technical calibration for evaluation of metabolic tumor volumes and variation in TLG (determined by mean SUV values and metabolic tumor volume) was admitted to be the best predictor of response to NAT when used at baseline and 12 weeks from NAT completion [49].

Metser et al. assessed the correlation between radiomic features and pCR by using machine learning algorithms and found that the systematizer trained on pre-treatment PET-CT had an accuracy of 92.8% in predicting pCR to NAT in patients with LARC [50,51].

### 4.8. Positron Emission Tomography/Magnetic Resonance Imaging (PET/MRI)

Although there is a limited number of clinical studies regarding the role of PET/MRI in post-NAT rectal cancer evaluation, still it has been proven to be at least equivalent to PET/CT and to standalone MRI due to its accuracy in T and N staging [52].

The systematic review of Crimì et al. in 2021 revealed that the reported sensitivity of T staging in [18F] FDG PET/MRI was in the range of 94–100%, specificity 73–94%, and accuracy 92–100%. The reported sensitivity, specificity, and accuracy for N staging were 90–93%, 92–94%, and 42–92% accordingly [53].

Several studies confirm that PET/MRI may be used for rectal cancer restaging after NAT and to select patients for rectum-sparing surgical approaches with potentially higher accuracy if compared with PET/CT and MRI alone [54,55,56,57,58].

### 4.9. Endorectal Ultrasound (ERUS)

ERUS, although proven to be a valuable method for the initial staging of rectal cancer, is unfortunately not the best choice in tumor reassessment after treatment [46]. The ability of accurate evaluation of tumor response to NAT with ERUS is restricted by the effects of the chemoradiation: tumor fibrosis, necrosis and peritumoral inflammation caused by therapy can remarkably compromise the preciseness of staging. All these reactions may look sonographically indistinguishable from residual tumor and may reduce the ability to differentiate the five layers of the rectal wall and result in over-staging. In general, it seems that ERUS has a particular role in restaging rectal cancer after NAT, but still, some specific changes in the rectal wall and surrounding structures have to be taken into account to avoid false staging [3].

## 5. Therapy-Related Predictive Factors of CR

### 5.1. Modalities of Neoadjuvant Radiation Therapy

#### 5.1.1. External-Beam Radiation Therapy (EBRT)

EBRT is the primary radiation technique used for NAT. It delivers radiotherapy (RT) to the rectal wall harbouring the primary tumor as well as to the complete mesorectum to treat tumor deposits in it [9].

Two different schedules of preoperative RT are standards of care–short-course preoperative radiation therapy (SCPRT) and long-course preoperative radiation therapy (LCPRT):SCPRT also known as the 5 × 5 Gray (Gy) regimen, offers 5 daily doses of 5 Gy (total of 25 Gy) and is usually followed by radical resection within one week of completing RT (<10 days from the first radiation fraction). SCPRT with delayed surgery is also a useful alternative to conventional SCPRT with immediate surgery offering similar oncological outcomes and lower postoperative complications [3];LCPRT regimens deliver daily doses of RT in significantly smaller fractions (about 1.8–2 Gy) over a longer period of 25 days to 28 days. The total RT dose delivered by this regimen is 45 Gy to 54 Gy and seems to be biologically equivalent to the 25 Gy short-course regimen [59].

It is not possible to give a rigid definition of which T and N sub-stages require SCPRT or nCRT [3]. In deciding whether to use SCRT vs. long-course nCRT, multidisciplinary considerations must be made, including the possibility of long-term toxicity and the need for tumor downstaging prior to surgery [1].

Two randomized comparative studies of SCPRT and LCPRT have been performed so far. The first was a Polish study published in 2006, which compared 155 patients who underwent SCPRT and 157 who underwent LCPRT [60]. The pCR rate was 0.7% for the SCPRT group and 16.1% for the LCPRT group. There was no difference in the incidence of postoperative complications or late effects, and no significant difference in overall survival (OS), relapse-free survival, or local recurrence rates between the two groups. Another randomized comparative study of SCPRT and LCPRT in 163 patients was carried out in Australia in 2012 [61]. Alike to the Polish study, this research also found that the pCR rate was only 1% in the SCPRT group, compared with 15% in the LCPRT group, and there were no differences in the incidences of postoperative complications or late effects, or the OS or local recurrence rates. The results of these two comparative studies were very similar, suggesting that SCPRT may have potential as a treatment that provides local control equivalent to that achieved by LCPRT without increasing postoperative or late complications [62]. Considering the influence of time on the development of CR to therapy, it has been suggested that SCPRT followed by delayed assessment of response may result in similar rates of CR to the observed after LCPRT [8].

For the treatment of LARC, a radiation dose of 45–54 Gy is recommended in the National Comprehensive Cancer Network (NCCN) and European Society for Medical Oncology (ESMO) consensus guideline. Nevertheless, Burbach et al. in 2014 found that dose escalation above 60 Gy for LARC results in high pCR rates and acceptable early toxicity [63]. In addition, Appelt et al. in 2013 demonstrated a significant dose-response relationship for tumor regression after preoperative CRT for LARC for dose levels in the range of 50.4–70 Gy [64]. These findings suggest that over 50 Gy of RT may be clinically relevant with acceptable toxicity; however, no major prospective trials exploring doses over 50 Gy have been performed yet. Additional studies are needed to confirm the safety and efficacy of dose escalation [62].

#### 5.1.2. High Dose Endorectal Brachytherapy (HDRBT)

HDRBT offers the advantage of the direct delivery of higher doses of RT to the mural rectal tumor, minimizing skin and sphincter exposure [62]. During preoperative HDRBT, 26 Gy is delivered in four daily applications of 6.5 Gy, covering the clinical target volume [65]. The HDRBT effect is limited to a 2-cm radius from the primary tumor, so it provides limited treatment of the mesorectal lymph nodes, blood and lymphatic vessels [62]. In a recent systematic review of 22 studies by Buckley et al. in 2017, it was concluded that the pCR rate following preoperative HDREBT with CRT ranged between 18% and 31% (weighted mean rate—22.2%). After preoperative HDREBT alone, the pCR rate ranged between 10.4% and 27% (weighted mean rate—23.8%). Preoperative HDREBT either alone or in combination with nCRT may result in a better pCR [66].

#### 5.1.3. Contact X-ray Brachytherapy (Papillon)

Contact RT was initially described by Papillon et al. (introduced in 1974) as another method for the direct delivery of RT to the rectal wall using a rigid proctoscope and a specially designed RT machine [62]. Contact X-ray brachytherapy (CXB) uses low-energy (50 keV) X-rays which are deposited mainly on the surface of the tumor and penetrate only a few millimetres of tissue beneath the tumor. Therefore, exophytic tumors are more suitable for this procedure than deeply infiltrative tumors [62,67]. X-rays are applied straight to the tumor under direct vision, minimizing the chance of a geographic miss. The dose falls off rapidly–the 100% dose is prescribed at the surface and the dose falls to 60% at 5 mm depth [68]. CXB treats only a small volume, usually <5 cm^3^ of tissue, compared with EBRT which treats much larger volumes of tissue, of about 1000–1500 cm^3^. Therefore, very high doses of radiation (~30 Gy, but with a biological dose equivalent to 100 Gy) can be safely delivered at each treatment fraction with very little collateral damage to the normal tissues around the tumor [62,67]. The treatment is given three times (30 Gy × 3) every two weeks. This regimen allows the normal tissues to recover during the 2-week break [68,69]. Like HDRBT, there is minimal toxicity but also minimal, if any, activity within the mesorectum. This treatment strategy has been suggested for the management of early tumors by RT alone as a form of local therapy or as a neoadjuvant approach followed by resection [62].

In a single centre experience where all patients received combined CXB and EBCRT–an initial cCR was seen in 144 (72%) of 200 patients following CXB [67]. The benefits of CXB were confirmed in Phase III randomised controlled trial in 2004–cCR was greatly increased in patients who received EBRT and CXB treatment compared to EBRT alone (24% vs. 2%) [70].

### 5.2. Modalities of Neoadjuvant Chemoradiation Therapy (nCRT)

Concurrent chemotherapy (ChT) during neoadjuvant RT has the added benefit of improved tumor downstaging and local control compared with RT alone [71]. In multiple phase II trials of LARC patients treated with preoperative RT alone, the pCR rates were notably lower (4–13%) if compared to the studies in which patients were treated with combined CRT (9–31%). Several randomized trials have demonstrated the benefits of adding concurrent chemotherapy to SCPRT and LCPRT by increasing local sensitization and systemic control of the disease [2,72,73,74,75].

In the ESMO and NCCN consensus guideline, 5-FU based chemosensitizers are recommended with conventional RT for the treatment of LARC to increase sensitivity to radiation [62,72]. According to ESMO guidelines, continuous intravenous infusions of 5-fluorouracil (5-FU) or oral capecitabine during CRT are recommended [2]. The equivalence of infusional 5-FU and capecitabine (an oral agent converted in tissues to 5-FU) has been established by the NSABP R-04 randomized controlled trial concerning rates of pCR, surgical downstaging, and sphincter preservation [2,73].

Several studies have investigated the usefulness of additional drugs with 5-FU-based chemosensitizers to improve the response rate. The largest number of prospective phase III randomized studies have tested regimens that incorporate oxaliplatin (ACCORD, CAO/ARO-04, STAR-01, FOWARC and NSABP R04 trials), showing controversial effects on the difference in pCR rates (no difference in pCR in STAR-01, NSABP R-04 and ACCORD 12/0405-Prodige 2 trial; higher rates of pCR in the group receiving oxaliplatin 17% vs. 13% in CAO/ARO/AIO-04 trial) [2,72]. Although most of these trials found that there was minimal or no difference in response rate between the two groups, acute toxicity and adverse events were significantly more common in patients who also received oxaliplatin [1,62]. Hence, the decision about adding oxaliplatin should be risk-balanced, taking into account the predicted toxicity for a particular patient [3].

Another promising radiosensitizer CPT-11 (camptothecin-11, irinotecan) has been evaluated in several published phase II trials, noting the usefulness of adding CPT-11 to CRT and reporting a pCR rate of 25–34% (Mehta et al., 2003; Choi et al., 2008). Although on the contrary some studies (Gollins et al., 2011; Mohiuddin et al., 2013) revealed that there was no significant difference between the treatment in terms of pCR or downstaging, an increased rate of acute toxicity was reported in the irinotecan group. Further studies are needed to confirm the usefulness of CPT-11 as a radiosensitizer [72,76].

Epidermal growth factor receptor (EGFR) inhibitors like panitumumab and cetuximab are already approved for the treatment of RAS wild-type metastatic colorectal cancer, but their role in LARC remains unclear. The results of adding cetuximab to 5-FU–based CRT regimens were disappointing–a pooled analysis of available studies indicate a pCR rate of less than 10% for combination therapy compared with 15% to 30% for standard 5-FU regimens and unacceptable toxicity as well. Panitumumab—only a few phase II trials have been published and the authors concluded that the addition of panitumumab to nCRT did not achieve the primary endpoint of expected pCR with additionally high toxicity [72,76]. Consequently, there is currently no role for the addition of EGFR-targeted therapy as a radiosensitizer in the treatment of LARC [76].

Anti-angiogenesis therapy with bevacizumab and sorafenib have not been approved themselves yet as well. Bevacizumab in combination with standard CRT has shown potential for rectal cancer although was not giving the expected increase in pCR. Sorafenib was giving encouraging results but still limited in small cohorts and phase I studies [76].

Poly (ADP-ribose) polymerase inhibitors (PARPs) like veliparib plus capecitabine-based CRT are demonstrating a pCR rate of 28%, although this class of potential radiosensitizer remains an area of interest and future studies are needed to clarify its role in rectal cancer [76].

With a clear focus of research on optimizing NAT, several novel modalities have been investigated–immunotherapy agents, cyclo-oxygenase 2 inhibitors, prostaglandin E2 receptor inhibitors, Ad3/Ad11p chimeric adenoviruses and nanoparticles etc. Despite extensive research and promising preclinical studies, a definite further agent in addition to fluoropyrimidines that consistently improves response rate has yet to be found [76].

Since combining drugs during nCRT regimens has failed to improve pCR rates, different schedules for the delivery of ChT in the neoadjuvant setting have been investigated [9]. The twofold rationale (known as well as “total neoadjuvant therapy”) for giving neoadjuvant ChT sequentially, either before or after nCRT, followed by surgery, was derived to improve the response of the primary tumor and to reduce the distant metastasis rate [76].

Additional chemotherapy before the start of standard nCRT is known as “induction chemotherapy”, however, no randomized comparative clinical trial has demonstrated that induction chemotherapy significantly improved the pCR rate, and the increased toxicity and even mortality associated with this treatment strategy has limited its widespread adoption and data acquisition till 2020 when new trials appeared [9,62].

Another delivery modification of ChT, known as “consolidation chemotherapy”, that includes standard nCRT followed by ChT during the “resting period” between nCRT and surgery, has yielded more promising results. High CR rates of up to 65% have been reported compared with historical controls of nearly 30% with more conventional nCRT regimens [9]. This regimen is attracting attention as a method of making use of the period between nCRT and surgery. As a longer waiting period after nCRT increases the pCR rate, there has been a tendency to extend this period in recent years. In some cases, it may exceed 2 months, and the goal of consolidation chemotherapy is to improve the pCR rate by the addition of FOLFOX or another chemotherapy regimen during this long waiting period [62]. The TIMING (Timing of Rectal Cancer Response to Chemoradiation) trial was a multicenter prospective phase II study examing the effect of adding 0, 2, 4 or 6 cycles of consolidation FOLFOX after nCRT in stage II-III LARC patients–pCR rates were significantly increased from 18% in the standard therapy group to 38% in the group receiving 6 cycles of FOLFOX with no difference in adverse events [77].

### 5.3. Neoadjuvant ChT

Neoadjuvant ChT alone using a fluoropyrimidine and oxaliplatin or combined with targeted agents has been proposed instead of nCRT in cT3 tumors, not threatening the CRM and cT4 tumors in the mid- and upper- rectum, to promptly treat potential micrometastases and individualising treatment options [3]. The FOWARC phase III study randomized 495 patients with LARC to either standard nCRT using concurrent 5-FU, nCRT with concurrent 5-FU and oxaliplatin, and FOLFOX chemotherapy alone. Although tumor downstaging was comparable between the standard nCRT and chemotherapy-alone arms (37.1% and 35.5%), the pCR rate was inferior with chemotherapy alone (14% vs. 6.6%) [78]. It was reported recently that there was no difference in disease-free survival (DFS) or OS between the three arms [79]. At present, there is more evidence to support the replacement of nCRT with chemotherapy using DFS as the primary endpoint, than for a cCR/organ preservation endpoint [76]. Hence, at this moment neoadjuvant ChT alone is not recommended for the treatment of localised, non-metastatic disease [3].

### 5.4. Interval to the Surgery

The optimal timing of surgical resection of LARC after nCRT or SCPRT remains controversial and is studied in trials. The ideal interval requires a balance between sufficient time after the RT for the maximal effects to be fully expressed (but before tumor repopulation) and the acute tissue reaction so that surgery can be carried out safely [3].

Tumor response to nCRT is time-dependant, sometimes taking months to achieve maximal tumor regression. Traditionally, the recommended interval between completion of NAT to surgery is 6–8 weeks–a timeframe that promotes tissue response and recovery from radiation and prevents the development of radiation-associated tissue fibrosis [2].

In a practice, there is a wide variation in the timing of surgery (4–12 weeks) due to recovery from treatment, patient/surgeon choice and waiting list issues [3].

There is an ongoing discussion on the best interval between treatment ending and response assessment. In the context of the “W&W” strategy, the aim is finding the perfect balance between the greatest tumor regression, therefore increasing the patient’s chance of being eligible for a “W&W” approach, while providing a safe and successful surgery, if surgery is inevitable. Several retrospective studies have suggested a higher rate of pCR when delaying surgery after nCRT [25].

In the case of SCPRT in resectable cancers, where downstaging is not required, “immediate” surgery is recommended to take place within 7 days from the end of NAT [3]. The Dutch trial (Marijnen et al., 2001) confirmed that preoperative SCRT also does not significantly downstage the tumor if the radiation-to-surgery interval is less than 10 days [80]. In case of SCPRT followed by “immediate” surgery (within 1–2 weeks) and “delayed” surgery (5–13 weeks)–earlier yp histopathologic stages and tumor categories, higher rates of pCR (11.8% vs. 1.7%) and higher rates of Dworak TRG4 (10.1% vs. 1.7%) were observed in case of “delayed” surgery [81].

If organ preservation is the goal–LCPRT (or SCPRT in non-fit patients) with a delay of 6 weeks until the first evaluation of tumor response is recommended. If there is no good response, surgery should be performed within 2 weeks. In case if cCR or near-cCR is achieved, restaging should be carried out after another 6 weeks, at which time the decision could be made whether or not to implement the “W&W” strategy [75]. Currently, the interval from the end of nCRT to surgery is based on the Lyon R 90-01 trial. This trial demonstrated that preoperative RT increased the rate of a pCR or near pCR from 10.3% at 2-week intervals to 26% at 6- to 8-week intervals. Thus far, the optimal interval is considered to be 6–8 weeks. The rationale for this interval is that it is expected to increase the pCR rate and reduce postoperative complications [82,83,84].

As to date, there is no final consensus regarding the interval between the end of nCRT and time to surgery, due to the promising results of clinical response in case of extension of the waiting period between NAT and surgery, and there is an observable trend in clinical trials to make careful delay of the surgery.

Because radiation-induced necrosis requires time to develop, a prolonged interval between radiation and surgery potentially increases pCR occurrence. In the study of Kleiman et al. in 2015, it was concluded that a radiation-surgery interval >8 weeks was associated with high pCR rates [85]. A meta-analysis incorporating 3584 patients by Petrelli et al. in 2016 reported an increase in pCR rates from 13.5% for an interval of 6–8 weeks to 19.5% for intervals longer than 8 weeks [86]. The observation that the post-nCRT lymph node positivity rate of 12% declines to less than 5% after an 8-week waiting period also supports the value of a longer waiting time [9].

However, the GRECCAR-6 trial revealed no significant difference between long (11 weeks) and short intervals (7 weeks) concerning pCR occurrence, although greater complications and difficulties in surgery were observed for participants with an 11-week interval [87].

Controversial results were also observed by Sun et al. who assessed the US National Cancer Database (NCDB) to answer the question of optimal timing. The overview included 11,760 patients with stage II-III rectal adenocarcinoma, treated between 2006 and 2012, who received nCRT followed by surgery. The authors found out that tumor downstaging increased during the waiting period, but when passed 56 days (8 weeks), there was no added benefit of delaying the surgery [88].

On the opposite, a retrospective review of more than 17,000 rectal cancer patients from the National Cancer Database found the optimal waiting period was 10–11 weeks, with a 27% greater odds of pCR for this interval compared to an interval of surgery of 6–8 weeks [2].

A comprehensive meta-analysis and systematic review was conducted by Donlin et al. in 2018—13 studies involving 19,652 patients were included. The meta-analysis demonstrated that pCR was significantly increased in patients with LARC and a waiting interval of ≥8 weeks between preoperative nCRT and surgery compared to a waiting interval <8 weeks or waiting interval of >8 weeks compared to ≤8 weeks. There were no significant differences in operative time, OS, DFS, the incidence of local recurrence, postoperative complications or sphincter preserving surgery. This study revealed that performing surgery after a waiting interval of 8 weeks after the end of preoperative nCRT is safe and efficacious for patients with LARC, significantly improving pCR without increasing operative time or incidence of postoperative complications [89].

In a British study by Evans et al. in. 2016, including patients considered to have a locally advanced disease–tumor downstaging recorded with MRI was higher in the group of patients had waited for 12 weeks rather than 6 weeks (58% vs. 43%), as were the rates of pCRs (20% vs. 9%) [90].

A further multicentre study by Figueiredo et al. in 2018 investigated outcomes for rectal cancer patients treated with surgery over 12 weeks after completing NAT (*n* = 76). Histopathological analysis of the resected surgical specimens demonstrated a pCR rate of 8.3% for those undergoing surgery within 12 weeks and 15.8% for those with extended interval to the surgery. There were no significant differences found regarding morbidity and mortality in either group [91].

In a study by Sloothak et al. in 2013, the pCR rate was significantly higher in patients with an interval of 15–16 weeks between CRT and surgery (18%) compared with the other time intervals (10.3% for less than 13 weeks, 13.1% for 13–14 weeks and 11.8% for more than 16 weeks respectively). These data suggested that delaying surgery until the 15th or 16th week after the start of nCRT (week 10 and 11 after a 5-week nCRT regimen) results in the highest chance of a pCR in patients with rectal cancer [92].

Another study by Garcia-Aguilar et al. in 2015 included patients with nCRT regimens and progressively longer interval periods before surgery. Although this was not a randomized study, patients in different groups were comparable. It was concluded that patients undergoing surgery after 12 weeks developed similar postoperative complication rates when compared with the standard 6-week interval. The study then kept on recruiting patients for progressively longer intervals: 6, 12, 18, and 24 weeks between nCRT and surgery. Even though additional systemic chemotherapy was offered to patients undergoing surgery after longer interval periods, delaying surgical resection to 20 weeks resulted in significantly higher pCR rates, with no negative impact on postoperative morbidity [93].

## 6. Host Related Predictive Factors of CR

### 6.1. Clinical Parameters

There are many characteristics of the patient that could potentially affect the outcome of the treatment. Examples include the presence of chronic disorders such as inflammatory bowel disease or diabetes and active smoking [25]. No-smoking history is found to be a clinical predictor for pCR in rectal cancer patients treated with long course nCRT [94]. Medication can be an important influencing factor, e.g., immunosuppressor agents like cyclosporin, tacrolimus or even simple corticosteroids. Currently, there is not much data about the interaction of medication and response to the treatment, but studies have found that the use of statins increases tumor regression with CRT [25]. As well–young age is a reported predictive factor of lower pCR rate following NAT [95].

### 6.2. Genetic Predisposition

Several specific genomic alterations associated with treatment response have been identified. Nine single-nucleotide polymorphisms (SNPs) were found to be associated with a better response to neoadjuvant CRT. For example, coronin 2A (CORO2A) rs1985859 was associated with a positive histopathologic response to CRT. Patients that are homozygous C/C genotype in MTHFR gene (rs1801133) were also found to be more predisposed to respond to CRT. Likewise, SNPs in the genes associated with miRNA processing (rs744910, rs745103 and rs1722821 for SMAD3; rs10719 for DROSHA; rs6088619 for TRBP) have also been found to be significantly associated with nCRT response in LARC patients [2].

## 7. Tumor Related Predictive Factors of CR

### 7.1. Clinical Parameters

Tumor size (≥3 cm), volume, tumor circumferential extent >60%, higher pre-treatment T stage is associated with highly aggressive tumor behaviour, indicating lower sensitivity to nCRT and lower rates of downstaging [96,97,98]. Tumor distance from the anal verge is strongly related to the clinical response as well. A retrospective analysis of Das P. et al. in 2007 (*n =* 562) revealed that the distance from the anal verge >5 cm was associated with significantly lower downstaging rates [96]. In a study by Bitterman et al. in 2015 (*n =* 138), the tumor distance ≥3 cm from the anal verge was found to be an independent predictor of lower CR rates [97]. In a prospective study of Patel et al. in 2016 (*n =* 827) where pCR was reached by 20% of the patients, it was found that pCR rates were 11% for tumors <4 cm, 24% for tumors 4–6 cm, 30% for tumors at 6–8 cm, 17% for tumors 8–10 cm, and 14% for tumors >10 cm from the anal verge. Patients with low tumors (<4 cm) and higher tumors (>8 cm), were less likely to have a pCR [99].

Clinically node-positive disease at the moment of diagnosis is an independent predictor of clinical response and is associated with decreased chance of achieving CR. It is possible that clinical node positivity may be a marker for a more aggressive disease that is less sensitive to local therapy [97,100]. pCR rates in clinically node-negative diseases are found to be three times higher than in node-positive diseases [101].

### 7.2. Morphological and Immunohistochemical Parameters

The presence of mucinous histology, poor tumor differentiation, macroscopic ulceration is related to decreased response rates [102,103,104]. Additional morphological parameters in untreated patients that play the role in overall prognosis and in the possibility to reach CR are–tumor budding and tumor-stroma ratio (TSR). The role of tumor budding (single tumor cells and small groups of tumor cells) is associated with a lack of response to NAT and a poor outcome in general as tumor buds are considered to be a feature of epithelial-mesenchymal transition and it is associated with the mesenchymal subtype. The role of TSR as a prognostic factor–the presence of extensive stroma between groups of tumor cells is associated with a poor outcome [26]. The relation between mucin pool formation also described as a colloid response, and NAT has been known for a long time as well. In patients with a pCR, approximately 27% of patients present with acellular mucin pool, but it is found that in pCR patients it does not affect prognosis [34].

In a review of Dayde et al. in 2017 extensive overview of tumor biomarkers was carried out. Expression of different proteins has been associated with response to nCRT, including vascular endothelial growth factor (VEGF), epidermal growth factor receptor (EGFR), p21, B-cell CLL/lymphoma 2 (Bcl2), BCL2-associated X protein (Bax), a marker of proliferation Ki-67 (ki-67), p53, hypoxia-inducible factor 1-α (HIF1-α), cyclooxygenase-2 (COX-2), E-cadherin, thymidylate synthase, matrix metalloproteinase-9 (MMP-9) and matrix metalloproteinase-2 (MMP-2). Protein biomarkers in tissues have been widely investigated and newly identified protein biomarkers are listed: ataxia telangiectasia mutated (ATM), meiotic recombination 11 homolog A (MRE11), X-ray repair cross-complementing protein 2 (XRCC2), cell cycle (polo-like kinase 1 (Plk1), PCNA-associated factor 15 (Paf15), c-MYC and proliferating cell nuclear antigen (PCNA), vaccinia-related kinase-1 and -2 (VRK1 and VRK2), focal adhesion kinase (FAK), golgi phosphoprotein 3 (GOLPH3), nuclear factor-κB (NF-κB), fibroblast growth factor receptor 4 (FGFR4), apoptotic protease-activating factor 1 (APAF-1) and COX2, survivin, Plectin-1 (PLEC1), Beclin 1 and desmoglein 3 (DSG3), transgelin (TAGLN), vascular non-inflammatory molecule 1 (VNN1), transketolase (TKT) and hydroxyacyl-CoA dehydrogenase (HADHA), 17-β-hydroxysteroid dehydrogenase type 2 (HSD17B2), 3-hydroxy-3-methylglutaryl coenzyme A synthase (HMGCS2) [105].

A study by Linders D et al. in 2021 presented tumor-targeted near-infrared (NIR) fluorescence imaging as a potential tool to improve response evaluation. The method allows real-time optical imaging by selectively highlighting cells that express certain molecular targets. To investigate the applicability of these targets, analysis of protein expression by immunohistochemistry in the tissue of rectal cancer patients with a pCR was performed. Promising targets in rectal cancer included carcinoembryonic antigen-related cell adhesion molecule 5 (CEACAM5, referred to CEA), epithelial cell adhesion molecule (EpCAM), urokinase-type plasminogen activator receptor (uPAR) and αvβ6 integrin. The immunohistochemical evaluation showed that EpCAM and CEA could be suitable targets for response evaluation after NAT, since the expression of these targets in the primary tumor bed was low compared with the diagnostic biopsy and adjacent pre-existent rectal mucosa in more than 90% of patients with a pCR [106].

Although several molecular biomarkers have been proposed as predictive of response to nCRT, none of these has reached the regular clinical application yet [107].

The comparison of the tumor immune microenvironment may also offer insight into the predicted response to NAT. Tumor-infiltrating lymphocytes (TILs)–low stromal Foxp3+ cell density is significantly associated with a good response to neoadjuvant CRT. Low PD-L1 expression both before and after nCRT is a negative prognostic marker, while another study demonstrated that high PD-L1 expression after nCRT was associated with vascular invasion, tumor recurrence and poor recurrence-free and OS [2].

A very informative tool for evaluation of possible clinical response is “immunoscore” (IS)–the combination of CD3+ and CD8+ T-cell densities in the tumor core and its invasive margin [101]. The IS is the first biomarker recommended by academic institutions quantifying the tumor immune infiltrate for a prognostic purpose (the ESMO guidelines 2020 and the 5th edition of WHO Digestive System Tumors) [107]. Further studies have proved that biopsies-adapted IS (IS_B_) is an even more promising prognostic tool. IS_B_ is a derivation of the IS performed in initial diagnostic biopsies before nCRT that has the advantage of evaluating the effect of the initial immune infiltrate (CD3+ and CD8+ T cells in the tumor) on response to nCRT and clinical outcome. Additionally, as nCRT induces histological and architectural changes, post-nCRT surgical specimen can not be assessed by the classical IS. IS_B_ was assessed in a multicentric cohort of 249 patients with LARC treated with nCRT followed by radical surgery. The IS_B_ levels correlated with the degree of histologic response to nCRT according to:the Dworak classification,the ypTNM staging, i.e., the post-surgical pathologic examination,the NAR score (Valentini V et al. in 2011 developed a nomogram for predicting local recurrence, distant metastases, and OS for patients with LARC. The nomogram for OS takes into consideration patient age, gender, the clinical tumor (cT) stage, pathologic tumor (pT) stage, pathologic nodal (pN) stage, the dose of radiotherapy, adjuvant chemotherapy administration and surgery type–abdominoperineal resection vs. low anterior resection) [107,108,109].

Patients with IS_B_ high were not found in the Dworak 0 non-responder group (no histologic response to nCRT) and most of the patients with IS_B_ low (80–90%) did not respond well to nCRT (no downstaging, Dworak 0, 1, or 2, or NAR low, or intermediate categories). IS_B_ combined with post-NAT imaging increased the accuracy of histologic good responders (ypTNM 0-I) prediction [107]. At the moment IS_B_ has several aspects of prognostic usefulness: it provides a strong and independent prognostic factor for DFS of patients with rectal cancer; it predicts the response to nCRT; IS_B_ combined with imaging post-nCRT discriminates the group of patients with a pCR to nCRT that should benefit from less invasive therapeutic strategies [108]. The clinical usefulness of the composite biomarker (imaging + IS_B_) was tested within a cohort of “W&W” patients (*n* = 73) with post-nCRT cCR (ycTNM 0). There was no evidence of relapse during the follow-up period in patients with IS_B_ high. These results suggest that IS_B_ could be a novel biomarker that might be used in the clinic for a better selection of patients eligible for the “W&W” strategy [107].

### 7.3. Tissue-Based Tumor Molecular Biomarkers

Although several molecular biomarkers in tumor tissues have been proposed as predictive of response to nCRT, they still are not used on an everyday basis.

Tumors that macroscopically look-alike can express a wide range of different mutations. There is intra-tumoral heterogeneity, which illustrates the danger of predicting pCR and supporting a non-operative management strategy that is based only on biopsy sample results. The Sao Paulo group in 2017 found that 60% of the mutations were present in only one fragment and only 27% of mutations were expressed in all fragments. The coexistence of cancer cell subpopulations within a single rectal cancer with distinct morphological features and genetic mutations makes the samples of single-biopsy not representative of the entire primary tumor. Unfortunately, a biopsy sample from one area of the primary tumor may contain cancer cells that are resistant to nCRT, while biopsy taken from another area—cells that are sensitive to nCRT [110].

Although there are some imperfections, molecular markers have the greatest future potential in the prediction of clinical response [111].

#### 7.3.1. DNA Mutation and DNA Methylation

KRAS, NRAS, and BRAF status are examples of attempts to predict pCR at a molecular level. KRAS mutation only and KRAS/TP53 mutation combination were found to be associated with a lower pCR rate in patients with LARC after nCRT in retrospective studies [12].

Duldulao et al. in 2013 carried out TP53 and KRAS genotyping in rectal cancer and presented that tumors with the KRAS mutation had a lower possibility to achieve pCR than those with wild-type KRAS. In their research, tumors with KRAS codon 13 mutations didn’t achieve pCR and also had a higher incidence of the TP53 mutation compared with tumors with other KRAS mutations. These results suggested that mutations in different KRAS codons may have different effects on the resistance of rectal cancer to nCRT and that the rectal cancers carrying TP53 and KRAS mutations have a lower opportunity to respond to nCRT compared with wild-type tumors [83,112].

Several researches have investigated the association of DNA methylation with response to nCRT and prognosis in LARC. While most investigations examined DNA methylation in a limited number of genes, Gaedcke et al. in 2014 profiled whole-genome methylation in 11 rectal cancer patients prior to nCRT using CpG island array analyses. Although the association of DNA methylation and response to nCRT was not evaluated in the study, the DNA methylation status of these regions was significantly associated with DFS [105,113].

#### 7.3.2. Gene Expression Profiles

Gene expression profiling of tumor tissues has the potential to identify gene signatures related to response to nCRT. Agosini et al. in 2015 examined gene expression profiles of pre-treatment biopsies. A set of 19 genes was significantly variously expressed between responders and non-responders. The resultant logistic regression model consisting of X-ray repair cross-complementing protein 3 (XRCC3), zinc Finger Protein 160 (ZNF160), additional sex combs-like protein 2 (ASXL2) and ATP dependent DNA helicase homolog (HFM1) successfully distinguished responders and non-responders with an accuracy of 95% [105,114]. The same group also identified seven genes (aldo-keto reductase family 1 member C3 (AKR1C3), C-X-C motif chemokine ligand 9 (CXCL9), CXCL10, CXCL11, matrix metalloproteinase-12 (MMP12), indoleamine 2,3-dioxygenase 1 (IDO1), and HLA class II histocompatibility antigen, DR α chain (HLA-DRA)) in immune system pathways, that can distinguish responders from non-responders [105,115].

Neuronal pentraxin II (NPTX2) has been also verified using quantitative real-time polymerase chain reaction in an independent set of tumor specimens from rectal cancer patients, and it has been found that decreased NPTX2 gene expression levels are associated with improved response to nCRT and prognosis [105,116].

#### 7.3.3. MicroRNA (miRNA)

Differential miRNA expression has also been associated with response to NAT. Several studies have revealed multiple miRNA’s that are upregulated or downregulated in rectal cancer if compared with normal mucosa, however, only part of them have the predictive potential for the response to NAT (Table 4) [2,12,14,105,117,118,119,120,121,122,123,124,125,126,127,128,129,130].

A review by De Palma et al. in 2020, who was assessing 61 articles, identified a total of 77 miRNAs that are holding a predictive value, however, only six miRNAs (let-7f, miR-21, miR-145, miR-622, miR-630, and miR-1183) exhibited significant differences in two or more independent studies [14].

### 7.4. Blood-Based Tumor Molecular Biomarkers

#### 7.4.1. Protein and Metabolites

Probst et al. in 2016 has investigated the relationship between pre-nCRT CEA levels and response to nCRT and OS in 18 113 LARC patients from a total number of 136,840 rectal cancer patients (data of National Cancer Data Base from 2006–2011). 47% of the patients had increased CEA levels before nCRT. Increased pre-nCRT CEA was independently associated with decreased pCR, reduced tumor downstaging, reduced pathological tumor regression and OS [2,105,131]. The association between elevated CEA levels and decreased response to nCRT has been reported in several studies and it has been found that elevated CEA level before CRT is associated with a decreased pCR rate and a low post-CRT CEA level <0.5 ng/dL is a significant predictor of a CR and improved OS and DFS, regardless of initial CEA levels [2,12,85,97,98,132].

Zhang et al. in 2015 assessed pre-treatment serum level of CEA and carbohydrate antigen 19-9 (CA19-9) in 303 LARC patients who received nCRT. While serum CEA levels were not significantly different in this study, increased serum CA19-9 levels were markedly correlated with poor OS, DFS and distant metastasis-free survival [105,133].

Other serum biomarkers—fibrinogen, carbonic anhydrase 9 (CAIX)–elevated levels before nCRT were found to be significant predictive factors for primary tumor regression, downstaging and pCR [105].

#### 7.4.2. MicroRNA (miRNA)

Measurements of the circulating miRNAs have been performed as an alternative to the biopsy-based tissue analysis. miRNAs are promising non-invasive biomarkers due to their presence in a variety of body fluids (found in plasma, breast milk, tears, bronchial lavage, amniotic, seminal, cerebrospinal, peritoneal and pleural fluids), their stability, simple detection and disease-specific expression in human tissues [14,134].

Currently, only a few studies have explored the potential of circulating miRNAs as predictive biomarkers in LARC patients [14].

D’Angelo et al. in 2016 performed miRNA microarrays analysis on biopsy specimen gathered before nCRT with a following evaluation of serum. Among eleven miRNAs which differed significantly between responders and non-responders (decreased in non-responders: miR-200a, miR-378, miR-33a, miR-338-3p, miR-30e; increased in non-responders: miR-299-5p, miR-125b, miR-409-3p, miR-127-3p, miR-154, miR-214), levels of miR-125b were additionally examined in serum from 34 LARC patients. It was found that serum miR-125b levels were notably higher in non-responders than responders [105,135].

Another study by Yu J et al. in 2016 reported that low miR-345 levels extracted from the serum of LARC responder patients (TRG 1–2) were associated with nCRT sensitivity when compared to non-responders. Similarly in a study by Hiyoshi Y et al. in 2017, low serum levels of miR-143 were associated with pathological response to nCRT in 94 patients [14].

Overall, currently known circulating miRNAs associated with response to NAT in LARC are: miR18b, miR-20a, miR-125b-1, miR-1183, miR-130a, miR-199b-5p, miR-301a-3p (high expression); miR-125b, miR-143, miR-100-5p, miR-345, miR-21-5p, miR-1246, miR-1229-5p, miR-96-5p (low expression). Although circulating miRNAs may reflect the tumor status, at the moment, this relationship has to be further investigated [14].

#### 7.4.3. Circulating Tumor Cells (CTCs)

Circulating tumor cells (CTC) may also present tumor response. Sun et al. identified CTCs by using epithelial cell-adhesion molecule (EpCAM) magnetic bead-based enrichment combined with a cytometric approach. A notable difference was detected in the levels of post-CRT CTCs and ∆%CTC (i.e., the percentage difference in CTC levels between pre-nCRT and post-nCRT) between responders and non-responders [105,136].

Magni et al. identified CTCs by using the CellSearch System in peripheral blood drawn before and after nCRT. Reduced amount of CTCs after nCRT were detected in the blood samples of responders, while no significant changes were noticed in the non-responder group [105,137].

#### 7.4.4. Circulating Cell-Free Nucleic Acids

Circulating cell-free nucleic acids (i.e., circulating DNA (ctDNA) or circulating RNA (ctRNA)), arising from primary and metastatic lesions, as well as CTCs, can be a potential material for liquid biopsy in rectal cancer patients. Sun et al. examined O6-methylguanine-DNA methyltransferase (MGMT) promoter methylation and KRAS mutation in plasma cell-free DNA (cfDNA) and detected that post-CRT cell-free DNA levels were found to be significantly lower in patients with response to CRT than in non-responders [138].

#### 7.4.5. Host Immune Response

Cytokines like interleukin (IL)-6 and IL-8 have been related with response to nCRT in rectal cancer [105,139]. Tada et al. in 2013 evaluated the concentration of monocyte chemoattractant protein-1 (MCP-1), interferon-gamma (IFN-*γ*), tumor necrosis factor-*α* (TNF-*α*), IL-2, IL-4, IL-6, IL-10, C-C motif chemokine ligand-5 (CCL-5), soluble CD40-ligand and TNF-related apoptosis-inducing ligand (TRAIL). While no significant association of cytokine levels with response to nCRT was observed before nCRT, levels of TNF-*α* and IL-6 after nCRT were significantly higher in non-responders compared to responders, also a significant reducement of CCL-5 and soluble CD40-ligand was detected in the responder group after nCRT [2,105,140].

Several researches have assessed the association of neutrophil-to-lymphocyte ratio (NLR) with response to nCRT [83,105,138]. Caputo et al. in 2016 assessed NLR and derived neutrophil-to-lymphocyte ratio (d-NLR) before and after nCRT in rectal cancer patients. Higher NLR and d-NLR after nCRT was significantly associated with poor response to nCRT and postoperative complications [105,141].

A high modified Glasgow prognostic score (mGPS; a combination of C-reactive protein and albumin levels), low lymphocyte-to-monocyte ratio and low platelet-to-lymphocyte ratio have been associated with a poor response to CRT as well [83].

## 8. Concluding Remarks

To this date, the accuracy of prediction and identification of pCR, using clinical, histological, endoscopic and radiological assessments, is not sufficient. Prediction of CR is complex, mainly due to the heterogeneity of patient and tumor factors, different treatment regimens (type, duration of chemoradiation, interval to the surgery) and assessment of treatment response. The difficulties of accurate identification of CR are related to the mismatch of the cCR and pCR in a particular proportion of the cases. The disconcordance of the pCR and cCR seems to be related to the sensitivity of the examinations that are used for setting up the diagnosis of CR.

At the moment, the diagnosis of cCR is based on clinical, endoscopic, MRI findings and the results of post-NAT biopsy and post-NAT CEA level. As it was assessed in the review, the clinical and endoscopic findings of local status are far enough not sufficient to predict the rate of pCR. Due to the different mechanisms of tumor mass reduction (shrinkage or fragmentation) and the possibility of geographical miss, the post-NAT tissue biopsy should not be considered as the affirmative examination for pCR as well. And the last–MRI examination, although it is potentially the most precise radiological method for evaluation of CR, can’t be assumed as the leading confirming parameter. Unfortunately, sensitivity and specificity levels prove that even with additional improvements (DWI-MRI) and in hands of an experienced radiologist, still there is a very high possibility for misdiagnosis. The imperfectness of previously mentioned diagnostic tools may explain the cases of local recurrence in patients who were diagnosed to have cCR and proposed for further “W&W” strategy. As several studies of postoperative morphological examinations have revealed, the numbers of “real” CR (pCR) are almost two to three times lower, therefore the existence of accurate diagnostic tools is crucial for safe guidance of “W&W” strategy. Implementation of new radiological approaches like PET/MRI may improve the overall accuracy of re-staging but at this moment it is still not used on an everyday basis.

At present, there are two main courses of development that are focused on the CR: (1) the exploration and evaluation of potentially more sensitive and specific diagnostic tools to reach more accurate results and reduce the disconcordance of cCR and pCR rates and (2) modification of the standard therapy approach to reach higher rates of pCR.

Multiple efforts of finding the clinical markers that would bring the pre- and post-NAT assessment to the closest point of LARC behaviour prediction are focused on winning the chance to avoid standard treatment approach in a particular population that would benefit from organ-preserving or non-surgical strategy with the same results of oncological safety and optimal prognosis. At present, such predictive markers are covering a wide area of sources, including clinical, morphological, genetical and molecular levels. Although several molecular biomarkers have been offered as potential predictors of response to nCRT, none of these has reached the regular clinical application yet.

Several programmed prognostic models have been offered to optimise the clinical evaluation and to increase the accuracy of pCR prediction. To this date, the performance of an artificial neural network (ANN) model evaluated by Huang et al. in 2020 in pCR prediction in patients with LARC has proved itself as the most accurate (if compared with k-nearest neighbour (KNN), support vector machine (SVM), naïve Bayes classifier (NBC) and multiple logistic regression (MLR) models) [98].

Another direction of the studies regarding CR is the optimisation and modification of current NAT to increase the numbers of pCR and long-term prognosis in LARC patients. At the moment, the standard NAT of LARC includes EBRT with or without concurrent ChT. Several studies reveal that increase of radiation doses, the addition of ChT agents are potentially promising for better tumor response and higher pCR rates. For example, the last update on NAT optimisation was revealed in the American Society of Clinical Oncology (ASCO) 2020 virtual scientific meeting where the results of clinical trials OPRA, RAPIDO and PRODIGE 23 were presented, establishing total neoadjuvant therapy as a new standard of care for LARC with promising short-term and long-term results [142]. However, in the unyielding way of finding the most effective therapy combinations, we shouldn’t forget the general principle–what is harmful to the tumor, most likely is harmful to the host as well. The risks of potential toxicity and NAT-related complications have to be carefully weighted before implementation in clinical practice to provide the most optimal treatment option to the patient and to avoid breaking the border of risks and benefits.

Another modality of the therapy modification involves the shifting of the surgical intervention after NAT with a tendency of careful delay. The most appropriate time window between the ending of the NAT and surgery is another aspect with not entirely clear defined boundaries. Studies suggest that the longer is the observational period, the higher the possibility of reaching pCR, but again–this strategy demands careful evaluation, risk and benefit weighting and strict patient cooperation.

In conclusion, the overall benefit in the determination of CR after NAT is the opportunity to guide more personalized treatment strategies. In clinical practice the presence of cCR demands more careful diagnostic approach, strict follow-up strategy, considering a certain risk of local recurrence or dissemination, but in the majority of the cases–is associated with relevantly higher patient overall life quality. Accordingly, there has been a growing interest in alternative approaches with less morbidity, including the organ-preserving “W and W” strategy, in which surgery is omitted in patients who have achieved a cCR. On the other hand, although surgical resection demands reconciliation with possible postoperative complications or permanent dysfunctions and is related with significant morbidity and decreased quality of life, the presence of postoperative pCR is associated with the most stable and stress-less oncologic outcome for the patient and medical professional.

The perseverance of clinical researches and implementation of new diagnostic tools and therapeutical algorithms hopefully will lead to the optimal, individualized and patient-friendly treatment solutions in a care of LARC patients.

## Figures and Tables

**Table 1 medicina-57-01044-t001:** Revised RECIST guideline (version 1.1).

Grade	Response Criteria
Complete response (CR)	The disappearance of all target lesions. Any pathological lymph nodes (whether target or non-target) must have a reduction in short axis to <10 mm.
Partial response (PR)	At least a 30% decrease in the sum of diameters of target lesions, taking as reference the baseline sum diameters.
Progressive disease (PD)	At least 20% increase in the sum of diameters of target lesions, the appearance of one or more new lesions is also considered progression.
Stable disease (SD)	Neither sufficient shrinkage to qualify for a partial response nor sufficient increase to qualify for progressive disease.

**Table 2 medicina-57-01044-t002:** Most common TRG classification systems.

TRG	Mandard	Dworak	Ryan/AJCC	MSKCC
TRG 0	-	no response	CR, no viable cancer cells	-
TRG 1	complete regression, no viable cancer cells, fibrosis extending through the different layers of the wall	minimal response (dominant tumor mass with obvious fibrosis, vasculopathy); fibrosis <25% of tumor mass	near-CR, single cells or rare small group of cancer cells	100% tumor response
TRG 2	rare residual cancer cells scattered through the fibrosis	moderate response (dominant fibrotic changes with a few easy-to-find tumor cells in groups); fibrosis 25–50% of tumor mass	partial response, residual cancer with evident tumor regression but more than single cells or rare small group of cancer cells	86–99% tumor response
TRG 3	increased number of residual cancer cells, fibrosis predominates	near CR (few microscopically difficult-to-find tumor cells in fibrotic tissue with or without mucous substance); fibrosis > 50% of tumor mass	poor or no response, extensive residual cancer with no evident tumor regression	≤85% tumor response
TRG 4	residual cancer outgrowing fibrosis	CR (no tumor cells, only fibrotic mass or acellular mucin pools)	-	-
TRG 5	absence of regressive changes	-	-	-

TRG—tumor regression grade; AJCC—American Joint Committee on Cancer; MSKCC—Memorial Sloan-Kettering Cancer Center; CR—complete response.

**Table 3 medicina-57-01044-t003:** Evaluation systems of rectal cancer response in post-NAT MRI.

Grade	Response	mrTRG (2012)	MERCURY (2012)	MERCURY (2016)	ESGAR (2016)
mrTRG 1	Complete	No evidence of treated tumor. Thin fibrosis, low-density signal on T2-weighted images with no evidence of intermediate signal intensity at the site of the treated disease.	No evidence of tumor signal intensity or fibrosis only	Linear/crescentic 1–2 mm scar in mucosa or submucosa only	Completely normalized rectal wall
mrTRG 2	Good	Dense fibrosis (>75%); no obvious residual tumor, signifying minimal residual disease, or no tumor. Dense fibrosis with no macroscopic evidence of intermediate T2 signal intensity.	Dense hypointense fibrosis with minimal residual tumor	No obvious residual tumor, signifying minimal residual disease or no tumor	Fibrotic wall thickening without clear mass
mrTRG 3	Moderate	>50% fibrosis or mucin and visible intermediate signal intensity. Predominating low signal fibrosis with macroscopic scattered or local intermediate signal intensity.	>50% fibrosis/mucin and visible tumor with intermediate signal intensity	>50% fibrosis/mucin and visible tumor with intermediate signal intensity	Residual mass (and/or focal high signal intensity on diffusion-weighted imaging)
mrTRG 4	Slight	little areas of fibrosis or mucin, but mostly tumor. Predominating intermediate T2-weighted signal with minimal or no fibrosis present.	Little areas of fibrosis/mucin, but mostly tumor	Little areas of fibrosis/mucin, but mostly tumor	
mrTRG 5	No response	Intermediate signal intensity; same appearance as that of the original tumor. Predominating intermediate T2-weighted signal with minimal or no fibrosis present.	Intermediate signal intensity, same appearances as original tumor/tumor regrowth	Intermediate signal intensity, same appearances as original tumor/tumor regrowth	

mrTRG—MR-modified Mandard grading system; MERCURY—Magnetic Resonance Imaging and Rectal Cancer European Equivalence Study; ESGAR—European Society of Gastrointestinal and Abdominal Radiology

**Table 4 medicina-57-01044-t004:** MicroRNA in rectal cancer associated with response to NAT.

Upregulated miRNA			Downregulated miRNA
miR-137	miR-143	miR-923	miR-720
miR-125	miR-194	miR-486-5p	miR-215
miR-1183	miR-866-3p	miR-34b	miR-190b
miR-483-5p	miR-379	miR-1274b	miR-29b-2
miR-125a-3p	miR-154	miR-450a	miR-590-5p
miR-1224-5p	miR-1542-5p	miR-450b-5p	miR-153
miR-622	miR-363	miR-99a	miR-519c-3p
miR-196b	miR-1290	miR-519b-3p	miR-561
miR-223	miR-188-5p	miR-1233	miR-30b
miR-494	miR-1471	miR-650	miR-145
miR-513a-5p	miR-1909	miR-1243	miR-148a
miR-513b	miR-21-5p	miR-125b	miR-375
miR-31	miR-671-5p	miR-345	miR-519b
miR-451	miR-630	let-7e	miR-1123
miR-335	miR-765		let-7f
miR-144	miR-193a-5p		miR-135b
miR-18a	miR-1290-3p		miR-16
miR-487a-3p	miR-382		miR-21
miR-1246	miR-19-3p		miR-200c

## Data Availability

Not applicable.

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
