# Peer review of "Evaluation and Predictive Factors of Complete Response in Rectal Cancer after Neoadjuvant Chemoradiation Therapy"

_medicina, 2021, doi:10.3390/medicina57101044_

Round 1

Reviewer 1 Report

I understand the importance of achieving CR from this study. The review is well-understood. Please improve it according to the comments.

1) The authors should emphasize the impact of the difference between clinical and pathological CR on clinical practice in patients with rectal cancer (e.g. affection to post-operative treatment, prognosis)

2) 7.3.3 Please make Table depending on the downregulation and upregulation of miRNA.

3) 7.4 Are any miRNA in blood examination related to that in tumor tissue?

4) Line 907-916.  I understand the present situation and the limitation of accuracy at each device. How do you think is the best modality for evaluation. Ultimately, do you think MRI is necessary? 

5) If some modalities such as MRI and PET-CT are combined for the evaluation, is accuracy for CR improved?

Reviewer 2 Report

In the abstract the authors have to specify what was the period of the research of literature.

The review has to be more structured. If this is a descriptive review the authors have to specify in the title but if this is a systematic review than all the article has to have another structure including the period of search, inclusion and exclusion criteria of the articles in this review

Please provide for all this informations "Rectal cancer represents approximately 35% of the total colorectal cancer incidence.This is a heterogeneous type of a cancer that attracts clinical attention due to the variety of treatment options. Most of the patients with early rectal cancer can be managed by surgery alone, however a significant proportion of patients present with a locally advanced disease that demands neoadjuvant therapy (NAT) with a purpose to reduce the local tumor burden and increase the safety and efficacy of further surgical treatment. NAT involves variety of treatment options including radiotherapy and chemotherapy, used alone or in combination. " the bibliographic support.

The discussion chapter has to be more poin to poin explained.

Round 2

Reviewer 1 Report

6.3.3. MicroRNA (miRNA) 

The authors list the lots of miRNA in Table.4. Are all miRNA related to the response to NAT? I think the authors should list the critical miRNA which was reported as a predictive factor for NAT. Some miRNA may not be related to the prognosis. 

Author Response

This manuscript is a resubmission of an earlier submission. The following is a list of the peer review reports and author responses from that submission.